# Pollen Sensitization Can Increase the Allergic Reaction to Non-Cross-Reactive Allergens in a Soy-Allergic Patient

**DOI:** 10.3390/ijerph20116045

**Published:** 2023-06-03

**Authors:** Daniela Briceno Noriega, Huub F. J. Savelkoul, Ad Jansen, Malgorzata Teodorowicz, Janneke Ruinemans-Koerts

**Affiliations:** 1Cell Biology and Immunology Group, Wageningen University and Research Centre, 6708 WD Wageningen, The Netherlands; 2Department of Otorhinolaryngology, Radboud University Nijmegen Medical Centre, 6525 GA Nijmegen, The Netherlands; 3Department of Clinical Chemistry and Hematology, Rijnstate Hospital, 6815 AD Arnhem, The Netherlands

**Keywords:** soy allergy, basophil activation, birch pollen season, seasonality, climate change and pollen sensitization

## Abstract

During and after the pollen season, an increase in food-triggered allergic symptoms has been observed in pollen-food syndrome patients, possibly due to seasonal boosting of pollen-IgE levels. It has been suggested that consumption of birch-pollen-related foods plays a role in seasonal allergenic inflammation. However, whether this increased pollen sensitization during the pollen season can also affect the allergenicity of allergens that are non-cross-reactive with birch pollen remains in question. This study presents the case of a patient with soy allergy and pollinosis, who experiences worsening of gastrointestinal (GI) symptoms during the birch pollen season even though the eliciting food factor does not cross-react with birch pollen allergens and their homologs (e.g., Bet v 1 and Gly m 4). The results showed a notable increase in sIgE for Gly m 4 (3.3 fold) and Bet v 1 (2.6 fold) during the birch pollen season compared to outside the birch pollen season, while Gly m 5 and Gly m 6 showed only a slight increase (1.5 fold). The basophil activation test (BAT) showed that in this patient Gly m 5 and Gly m 6 are clinically relevant soy allergens, which correlates with the reported clinical symptoms to processed soy. Moreover, the BAT against raw soy shows an increase in basophil activation during the birch pollen season and a negative basophil activation result outside the birch pollen season. Thus, the worsening of GI symptoms could possibly be due to an increase in IgE receptors, an over-reactive immune system, and/or significant intestinal allergic inflammation. This case highlights the importance of including allergens that do not cross-react with birch pollen and using a functional assay such as the BAT to evaluate clinical relevance when assessing birch pollen seasonal influence on soy allergenicity.

## 1. Introduction

Legume proteins share homology; however, they are not necessarily similarly allergenic. Presently, the most prevalent legume allergies are peanut, soy, lupine, chickpea, lentil, and pea [1,2]. In Europe, soy comprises one of the eight most common food allergens [3,4,5], with eight soybean allergens recognized by the WHO-IUIS Allergen Nomenclature Subcommittee [6]. Soy allergy is commonly characterized by IgE sensitization to the storage proteins Gly m 5 (vicilin, 7S globulin) and Gly m 6 (legumin, 11S globulin), which constitute 65% to 80% of the total seed protein and can provoke severe allergic reactions to all kinds of dietary soy, including processed foods [7,8,9]. Moreover, these allergens have been reported to be stable to heat and gastric digestion and may be associated with food allergies primarily associated with sensitization through the gastrointestinal tract [8]. In the Netherlands, most soy-allergic patients appear to be sensitized to Gly m 5 and Gly m 6, known as ‘conventional’ soy allergy, according to a recent report [3]. Another form of soy allergy is linked to Gly m 4 (a pathogenesis-related protein (PR-10)), which has been identified as the most common allergen in soy patients with birch pollen allergy [8,10]. Moreover, it is not uncommon that patients who are allergic to birch pollen show allergic symptoms after the ingestion of certain fruits and vegetables due to the cross-reactivity of IgE antibodies induced by sensitization to the major birch pollen allergen Bet v 1 with homologous food allergens, Gly m 4 in the case of soy allergy [11]. In contrast to ‘conventional’ soy allergy, Gly m 4 has an elevated sensitivity to heat and pepsin digestion; thus, Gly m 4 allergy seems to be associated with immediate reactions known as pollen-food syndrome (PFS), a clinical group of signs and symptoms which result from cross-reactivity between pollen and plant food allergens [12]. In the case of Gly m 4, PFS is linked to the consumption of unprocessed soy, often soymilk [10,13,14,15,16]. Furthermore, symptoms can be as severe as anaphylactic reactions, which occur in approximately 10% of birch-allergic patients who present a cross-allergic reaction to soy [17,18]. Soy-based drinks, such as soymilk, can also cause systemic allergic reactions, which suggests that drinks contain a high amount of soy protein that has not been thoroughly thermally processed; as liquids can partially bypass gastric digestion, they can thus quickly reach the intestine [13,17,19]. IgE-mediated food allergies, such as soy, may result from sensitization through the gastrointestinal tract or through a less well-recognized form via the respiratory tract, which results in primary sensitization to homologous pollen allergens, causing reactivity to cross-reactive food allergens [20]. An example of this form of sensitization is evidenced by the fact that in Europe, plant food allergy is strongly influenced by sensitization to birch pollen proteins [20,21]. Moreover, it has been reported that approximately 60% of food allergies in adolescents and adults are linked with an inhalant allergy [13]. This association is further supported by the fact that during and after the pollen season, an increase in food-triggered allergic symptoms has been observed in pollen-food syndrome patients, possibly due to seasonal boosting of pollen-IgE levels [15,22,23]. Nonetheless, clinical evidence in support of seasonal variation of serum IgE (sIgE) levels has been quite limited so far. The levels of birch pollen have not only risen in recent decades, but the period of exposure has increased due to climate change, which has resulted in a rise in the prevalence of birch pollen sensitization; further research on this topic is warranted [23]. In Northern Europe, birch is the major pollen-allergen-producing tree; the main flowering period starts at the end of March, with pollen values peaking in May, and the duration is remarkably temperature-dependent and can last as long as 8 weeks [24]. Moreover, birch has the greatest allergenic potential of the allergenic trees, with positive skin prick tests (SPTs) to birch allergens being observed in 5% of the Dutch population [23,24]. Additionally, a high rate of patients, 50% to 80%, with a respiratory allergy may be sensitized and clinically allergic to other allergens, known as polysensitization, with patients tending to gain sensitizations over time [25,26,27,28]. Polysensitization includes cross-reactivity, when the same IgE binds to several different allergens with common structural features, and co-sensitization, when different IgEs bind to allergens that may not necessarily have common structural features [28]. Additionally, Magnusson et al. reported that birch-pollen-allergic patients show signs of elevated numbers of eosinophils, IgE-positive cells, CD3+ T cells, and CD11c dendritic cells during the pollen season, consistent with an increased severity of allergic symptoms [29]. Furthermore, elevated IgE levels to most PR-10 proteins have been observed during the pollen season, particularly in patients, which indicates that cross-reactions between birch-pollen-related foods may play a role [30]. Previous studies have reported an increase in pollen sensitivity as the pollen season progresses [30,31,32,33,34]. Together, these findings support the hypothesis that during the birch pollen season, it is possible that the consumption of birch-pollen-related foods (e.g., soy) plays an important role in allergenic inflammation, which increases during the pollen season [29,30,31,34]. However, whether this increased pollen sensitization during the pollen season can also affect the allergenicity of allergens that are non-cross-reactive with birch pollen remains in question.

The aim of this study is to present the case of a soy-allergic patient with clinical symptoms after consumption of processed soy and whose symptoms increase during the birch pollen season, which suggests a pollen–soy-related allergy. The sIgE to the major birch allergen and soy allergens was measured in and outside the birch pollen season. Additionally, as sensitization does not always correlate to clinical symptoms, a basophil activation test (BAT) was used to evaluate the clinical relevance of sIgE sensitization to the major relevant soy allergens, Gly m 5 and Gly m 5, and assess the patient’s capacity to degranulate basophils outside, at the start of, and during the peak of the birch pollen season [35,36,37]. An ImmunoCAP Inhibition test was performed to confirm that the patient’s sIgE binds to processed soy allergens, since her symptoms occur upon consumption of processed soy products.

## 2. Case Presentation

A 26-year-old Dutch woman presented to the Allergy Centre at the Rijnstate Hospital in Arnhem, the Netherlands, with a chronic history of multiple food allergies, including a peanut allergy which begun at an early age, and recorded episodes of anaphylactic shock (AS). The patient developed PFS, abdominal pain, diarrhea, and AS after the ingestion of soy products; the patient indicates that she has never consumed soymilk or soymilk products.

Recently, she reported a difference in soy tolerance during and outside the birch pollen season; she is capable of tolerating small amounts of processed soy outside the birch pollen season without experiencing any symptoms but must maintain a strict diet during the birch pollen season (Table 1). During the birch pollen season, consumption of even small amounts of processed soy will lead to the development of PFS, gastrointestinal symptoms, and on two occasions, AS (Table 1).

Additionally, the patient reports a history of hay fever with moderate to severe symptoms, including sneezing, runny nose, itchy eyes, nose, and throat, and persistent asthma aggravated by exposure to furry animals and respiratory irritants.

In the general physical examination, discoid eczema patches were observed in the head and neck region. The rest of the physical examination was normal. The only medication prescribed is Budesonide/formoterol (Symbicort^®^) daily. The patient’s emergency medications are desloratadine, salmeterol, and prednisone and they carry an EpiPen.

## 3. SPT and sIgE Results

The SPTs showed a slightly positive reaction to soy, strong reactions to peanut, apple, and hazelnut, and weak reactions to celery, tomato, nutmeg, and pea. It has been reported that a patient’s sensitive skin or dermatographism can alter SPT results; thus, the results of the SPT to soy may have been influenced by the patient’s atopic eczema [38].

During the peak of the birch pollen season, the patient showed positive sIgE values against whole soybean and multiple soy allergens: Gly m 4, Gly m 5, and Gly m 6. The patient also showed a positive value to the major birch pollen allergen Bet v 1 (Table 1). Furthermore, sIgE to Gly m 4 and Bet v 1 notably increases, 3.3-fold and 2.6-fold, respectively, during the birch pollen season compared to the measured sIgE values out of the birch pollen season (Table 1). However, for sIgE for Gly m 5 and Gly m 6, a slight increase across the different seasons is observed, 1.5-fold for both allergens (Table 1).

Additionally, molecular allergy diagnostic was used to evaluate the patient’s sensitization to various inhalant allergens and peanut allergens; the results of the defined partial allergen diagnostics (DPA-Dx) showed that the patient was positive to Bet v 1 and negative to Bet v 2, Bet v 4, and Bet v 6, as shown in Table 2.

Because the patient had an unequivocal and convincing clinical history to soy plus a positive sIgE result in addition to a history of AS, an oral food challenge was deferred due to the high probability of a reaction [39].

## 4. Basophil Activation Test (BAT)

Since IgE sensitization does not equal the clinical manifestation of an allergy, a direct basophil activation test (BAT) was performed [37,38]. The BAT was performed on whole blood using the Flow CAST^®^ kit (Bühlmann, Basel, Switzerland) following the manufacturer instructions [40]. Basophil activation was expressed as the %CD63-positive basophils (CD63%+); the cut-off value for positive basophil activation was set at >15% CD63%+ [41,42]. The patient was instructed to stop any medication a week prior to blood collection. EDTA blood samples were freshly incubated with basophil stimulation buffer (BSB), with the negative control being BSB without allergen. The allergen concentrations used were 1, 10, 100, and 300 ng/mL for Gly m 5 as well as for Gly m 6 and 5, 10, 20, and 40 ng/mL for Ara h 2.

The BAT showed positive results during the pollen season for Gly m 5 and Gly m 6 (Figure 1A) as well as for the major peanut allergen Ara h 2 (Figure 1B). The positive BAT result for Ara h 2 correlates well with the clinical history of the patient, who describes a severe form of peanut allergy that started at an early age. The BAT results outside, at the beginning, and during the peak of the birch pollen season show a season-dependent change in basophil stimulation, remaining negative outside the birch pollen season, starting to rise at the beginning of the birch season, and increasing during the peak of the birch pollen season (Figure 1C). These results correlate with the medical history, where the patient reports that outside the birch pollen season consumption of small amounts of processed soy is possible without experiencing any symptoms. However, during the birch pollen season a strict diet must be maintained and consumption of even small amounts of processed soy can lead to clinical symptoms such as OAS, abdominal pain plus diarrhea, and on two occasions, even AS (Table 1).

## 5. ImmunoCAP Inhibition Assay

Since the patient’s complaints arose after the consumption of processed soy, an ImmunoCAP Inhibition test was performed to measure the binding capacity of sIgE to processed allergens [43]. Raw soy protein extract (SPE) was used as an inhibitor protein, two times the diluted serum with no inhibitor protein (0% inhibition) was used as the negative control, and changes to the IgE binding capacity were measured in raw soy and two forms of processed soy: (i) processed soy type 1 (raw soy protein extract heated at 121 °C without glucose for 10 min) and (ii) processed soy type 2 (raw soy protein extract heated at 121 °C with glucose for 10 min).

To evaluate which soy allergens were more susceptible to changes caused by processing techniques (heating and glycation), the inhibition of sIgE against Gly m 4, Gly m 5, and Gly m 6 was measured and compared. The final concentrations used were 1, 5, and 25 μg/mL; the sIgE levels were measured using the Phadia250^®^ instrument (Thermo Scientific, Germany), and the percentage of inhibition was calculated with the following formula:(1)$$\%\;Of\;inhibition=\frac{\left(IgEo\%-IgEx\%\right)}{IgEo\%}\times 100$$

The results showed no inhibition for Gly m 4 for raw SPE and the two forms of processed soy (Figure 2A); therefore, Gly m 4 was either not present or at a very low concentration in the soy extracts. For Gly m 5 and Gly m 6, a strong inhibition signal was observed (Figure 2B and Figure 2C, respectively), indicating that the positive BAT results with soy extract are due to the patient’s reactivity to Gly m 5 and Gly m 6.

## 6. Discussion

Presently, it has been reported that there is a high prevalence of sIgE to Gly m 4 in a Bet v 1-sensitized patient, which usually indicates that birch PFS may be clinically relevant after the ingestion of soy products, usually soymilk [14,15,17]. However, the present study presents the case of a patient mainly sensitized against soy storage proteins Gly m 5 and Gly m 6 (sIgE and BAT results), who does not consume soymilk or soymilk products but reacts to processed soy (ImmunoCAP Inhibition) and still experiences PFS plus gastrointestinal symptoms and a difference in soy tolerance in and out of the birch pollen season.

In the present study, a BAT against Gly m 4 was not performed; thus, the patient’s clinical reactivity to Gly m 4 was not analyzed. To our knowledge, a BAT with Gly m 4 has been reported only once and in a patient with Gly m 4-exclusive soy allergy, which resulted in no degranulation for the CD63 marker for the all the allergen concentrations used, except 67.5 ng/mL [8]. In the present study, no inhibition was observed for Gly m 4 for either raw SPE or the two forms of processed soy. Mittag et al. reported that the content of Gly m 4 in soy products is highly variable, ranging from 0 to 70 mg/kg and depends strongly on the degree of food processing [18]. Moreover, Gly m 4 degrades during heating, with Vissers et al. reporting that after 30 min to 4 h of heating, no Gly m 4 could be detected [44]. Thus, the allergenicity of Gly m 4 in soy products strongly depends on food processing conditions [18,44,45,46]. Therefore, the no-inhibition results observed in our study might be explained by the absence or by the very low concentration of Gly m 4 in the soy extracts. Moreover, the commercial extracts that are currently available generally contain very low amounts of Gly m 4, 0.01–0.1% in total soy crude protein soy extracts; thus, tests for allergen-specific IgE tend to be not sensitive enough in cases of suspected Gly m 4-induced soy allergy [45,46].

Outside and during the birch pollen season, the patient shows sIgE differences for Gly m 4 and Bet v 1 but not for Gly m 5 and Gly m 6. Previously, higher IgE sensitization rates to pollen during or following the pollen season compared to out of the pollen season have been reported [47,48,49]. The data relating to the current patient suggest that sIgE to birch pollen is 2.6-fold higher during the birch pollen season (Table 1), which is consistent with previously described results [49]. However, studies evaluating sIgE seasonal variation are limited and focused mostly on symptomatic aeroallergens [49]. The clinical significance of this change in sIgE values is still uncertain, although it has been suggested that higher sIgE levels to birch are associated with higher symptom frequency [23,49,50]. Furthermore, studies that explore the sensitization pattern of birch-pollen-related foods in relation to the birch pollen season are extremely limited and do so in patients with a well-established birch pollen–food-associated clinically relevant allergy [20,22,51,52]. As observed in the presented case, the sIgE values for Gly m 4 notably increase during the birch pollen season: 3.3-fold for Gly m 4 compared to 1.5-fold for Gly m 5 and Gly m 6.

The current hypothesis suggests that patients sensitized to birch pollen (Bet v 1), which cross-reacts with a wide range of fruits and vegetables (e.g., Gly m 4), can experience an increase in food allergy episodes during and immediately following the birch pollen season due to the seasonal boosting of pollen-IgE levels [14,15,22,29,30]. Rentzos et al. reported that birch-pollen-allergic patients have clear signs of an ongoing intestinal mucosal inflammation (eosinophil infiltration and increased numbers of IgE+ cells, mainly mast cells) which is aggravated during the pollen season [30]. This over-reactive immune system may be a reaction to the pollen itself rather than pollen-related food items [29,30,53]. Moreover, sensitized individuals produce more allergen-specific IgE antibodies; the binding of allergen to allergen-specific IgE antibodies on basophils leads to the cross-linking and activation of the high-affinity IgE receptor (FcεRI) and low-affinity IgE receptor (CD23) responsible for allergic inflammation [54]. Carlsson et al. reported a correlation between IgE bound on basophils with increasing IgE levels and increasing receptor expression [55]. The present BAT results show a different basophil activation dependent on the birch pollen season, with basophil activation starting to increase at the beginning of the birch pollen season and increasing significantly during the peak of the season. These BAT results correlate with the clinical history, which shows a difference in soy tolerance in and out of the birch pollen season. These BAT results suggest that the increase in pollen sensitization during the pollen season may increase the high- (FcεRI) and low-affinity (CD23) IgE receptors, thus increasing basophil activation.

The patient in the present case study shows a clinical history of different tolerance of processed soy outside and during the birch pollen season, while chronically avoiding consumption of soymilk and soymilk products. To obtain the many commercially available soy products, soy must undergo various processing steps, with conventional thermal methods being one of the most used processing techniques [56,57]. Higher heating temperatures lead to an altered protein structure, thereby increasing protein digestibility and quality [57]. If sugars are present during the heat treatment, a reaction known as glycation or the Maillard reaction occurs [56]. Thermal processing prompts conformational changes in food proteins, affecting not only their digestion and absorption but also their recognition by immune cells and binding to IgE antibodies and, thus, their allergenicity potential [56,58]. The results observed in the ImmunoCAP Inhibition test show high levels of inhibition for Gly m 5 and Gly m 6; both of these allergens, unlike Gly m 4, are thermo-stable and, thus, retain their allergenic potential [7]. Moreover, compared to raw SPE, a decrease in allergenicity is observed for Gly m 6 for the two types of processed soy, which added to the low levels of sIgE (Gly m 4 > Gly m 6 > Gly m 5), and the medical history may suggest a tolerance development towards soy proteins.

Despite the main limitations of a case report, mainly that generalization of the presented hypothesis to a broader population is not possible and there is a danger of result over-interpretation, the major advantage of a case study is the possibility of presenting novel and uncontrolled observations regarding clinical findings. To our knowledge, this is the first time that a case is presented that shows a marked increase in basophil activation during the pollen season in a soy-allergic patient whose food allergy is a primary dietary form linked to sensitization against soy storage proteins Gly m 5 and Gly m 6. Moreover, these results suggest that the change in soy tolerance during the birch pollen season was not only caused by Gly m 4 sIgE increase in the birch pollen season (as shown in Figure 3). Therefore, it is possible that the seasonal soy oral tolerance and worsening GI symptoms during the birch pollen season are due to an increase in IgE receptors, an over-reactive immune system, and/or significant intestinal allergic inflammation. Moreover, it would be expected that this immune system reaction is reversible since the patient experiences this change in diet restriction and symptomology every year. In future research, such patients should be followed for a long period of time (a minimum of two years) to evaluate if this phenomenon is indeed reversible and to what degree.

## 7. Conclusions

The case presented illustrates that it is crucial that clinicians become aware that it may not be only birch-pollen-associated soy allergy patients that can suffer an increase in gastrointestinal symptoms and changes in soy tolerance during the birch pollen season and, hence, adjust dietary advice accordingly.

Moreover, the changes in the clinical reactivity to soy according to seasonality measured by the BAT and reflected in the clinical history of the patient demonstrate the advantages of including a functional assay for monitoring allergic seasonal changes, which will be of particular importance since the effects of climate change influence allergic diseases [23,24,34,54]. Climate change not only increases the duration of exposure to aeroallergens, thus increasing allergic sensitization, but could also increase the level of allergenic airborne pollen and allergic symptoms, including those caused by non-pollen-related food allergens, as demonstrated by this case report [24,34]. Thus, this single case study highlights the importance of using seasonal sIgE measurements for birch plus food allergens and to include allergens that do not cross-react with birch pollen, in the case of soy-allergic patients, Gly m 5 and Gly m 6, in future research. Additionally, these studies should include not only seasonal measurements of allergic markers such as sIgE and symptomatology but also add a functional assay such as the BAT to evaluate clinical relevance.

## Figures and Tables

**Figure 1 ijerph-20-06045-f001:**
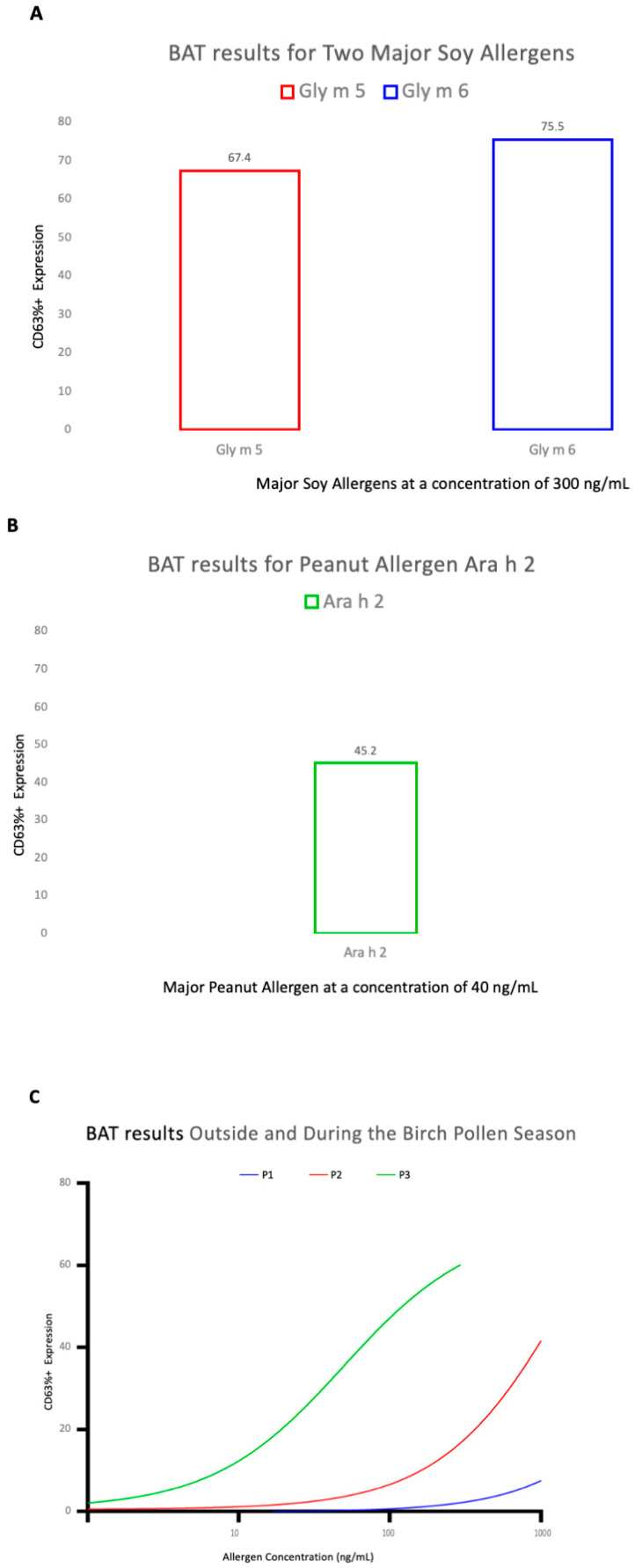
BAT results for soy and peanut allergens plus measurements outside, at the start, and in the peak of the birch pollen season. (**A**) BAT results for soy allergens Gly m 5 and Gly m 6 at an allergen concentration of 300 ng/mL; (**B**) BAT results for peanut allergen Ara h 2 at an allergen concentration of 40 ng/mL; (**C**) BAT results against raw soy extract at an allergen concentration of 300 ng/mL. Period 1: December, outside of the birch pollen season (P1—blue color), Period 2: March, at the start of the birch pollen season (P2—red color), and Period 3: May, at the peak of the birch pollen season (P3—green color).

**Figure 2 ijerph-20-06045-f002:**
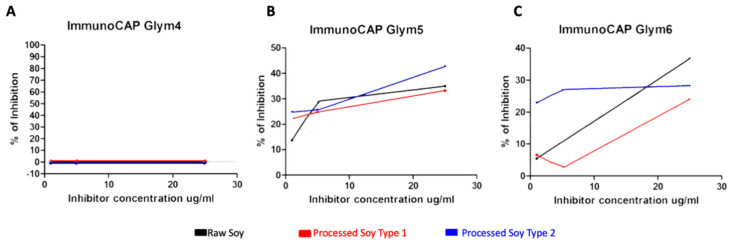
ImmunoCAP Inhibition test results for raw soy extract, processed soy type 1, and processed soy type 2. (**A**) Percentage of Inhibition against Gly m 4; (**B**) Percentage of Inhibition against Gly m 5; and (**C**) Percentage of Inhibition against Gly m 6.

**Figure 3 ijerph-20-06045-f003:**
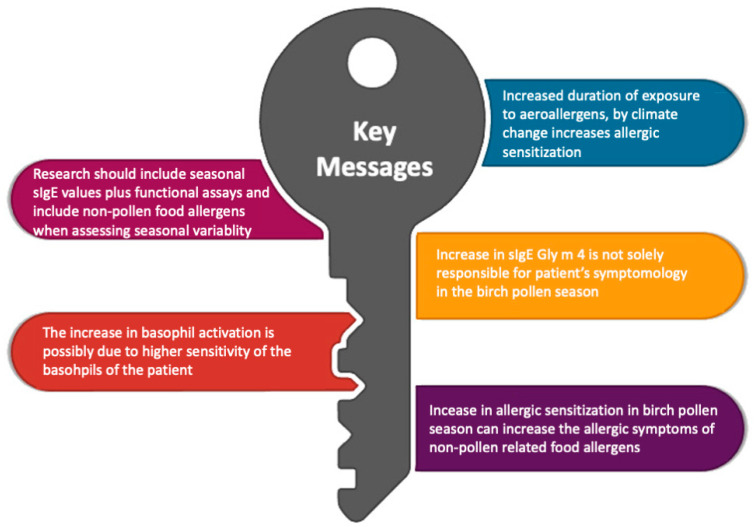
Key Messages.

**Table 1 ijerph-20-06045-t001:** Patient sIgE values measured outside and during the peak of the birch pollen season plus soy diet restrictions and reported patient symptoms.

	ImmunoCAP IgE (kU/L) *					Soy Diet	Symptoms
	Soybean	Gly m 4	Gly m 5	Gly m 6	Bet v 1		
Outside of the Birch Pollen Season	0.63	2.13	0.91	1.42	6.3	Tolerance to processed soy	-
Peak of the Birch Pollen Season	1.15	7.12	1.33	2.14	16.3	Strict diet	OAS, AS, AP, D

* Specific IgE (sIgE) values tested using the ImmunoCAP (Thermo Fisher Scientific^®^, Waltham, MA, USA); manufacturer’s recommendations were followed, with a positive result when IgE levels were ≥0.35 kU/L; OAS = oral allergy symptoms; AS = anaphylactic shock; AP = abdominal pain; D = diarrhea.

**Table 2 ijerph-20-06045-t002:** Birch and peanut allergens measured by DPA-Dx.

DPA-Dx (Ku/L)			
	kU/L *		kU/L *
Birch	38.5	Peanut	98.1
rBet v 1	43.8	rAra h 1	0.7
rBet v 2	<0.35	rAra h 2	11.4
rBet v 4	<0.35	rAra h 3	2
rBet v 6	<0.35	rAra h 6	37.2
		rAra h 7	21.1
		rAra h 5	<0.35
		rAra h 9	<0.35

* Cut off for positive values > 0.35 kU/L.

## Data Availability

The data presented in this study is available on request from the corresponding author. The data is not publicly available due to privacy.

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
