# Peer review of "Pollen Sensitization Can Increase the Allergic Reaction to Non-Cross-Reactive Allergens in a Soy-Allergic Patient"

_ijerph, 2023, doi:10.3390/ijerph20116045_

Round 1
Reviewer 1 Report
Comment #1: There is a difference between oral-allergy syndrome (OAS) and pollen-food allergy syndrome (PFS). The term OAS is used as a descriptor for the particular oropharyngeal symptoms elicited following ingestion of an implicated food and PFS is used for the clinical entity of OAS resulting from cross-reactive allergens between pollen and plant foods. See doi: 10.1097/ACI.0b013e32833973fb. Please change OAS to PFS throughout the manuscript.
Comment #2: Why wasn’t a SPT to birch pollen performed? Also, can the authors explain why the SPT result to soy was only slightly positive? Especially because the patient has had a history of anaphylactic shock and positive sIgE values to soy. Did the authors consider performing oral food challenges to soy for this patient? Please also provide the SPT value – how many mms?
Comment #3: It is unclear what the authors meant by ‘birch pollen-associated clinically relevant soy allergic’. A clear definition of this should be provided in the main text. Please present the patient’s test results and symptoms against this definition. At the moment, it is a bit confusing as many different tests and symptoms were performed/reported, and at different times.
Comment #4: The patient is also peanut allergic. Did the authors check if the patient was also sensitised to peanut allergens that are not cross-reactive to birch pollen?
Author Response
Dear Reviewer:
We would like to thank you for your comments and suggestions; we have revised the manuscript accordingly and hopefully to your satisfaction. Please note that the indicated lines below are for the no markup version on Word, since the version with all mark-up can alter the line numbering.
Comment #1: There is a difference between oral-allergy syndrome (OAS) and pollen-food allergy syndrome (PFS). The term OAS is used as a descriptor for the particular oropharyngeal symptoms elicited following ingestion of an implicated food and PFS is used for the clinical entity of OAS resulting from cross-reactive allergens between pollen and plant foods. See doi: 10.1097/ACI.0b013e32833973fb. Please change OAS to PFS throughout the manuscript.
Answer: We would like to thank the reviewer for pointing out this definition discrepancy in the manuscript, we have now corrected this issue, defined the term pollen-food allergy syndrome more clearly (line 62-63) and added the suggested reference (ref# 12). In table 1, the OAS term was used to describe oropharyngeal symptoms in the questionnaire filled by patients; therefore, it remained as such and explained in the legend of the table.
Comment #2: Why wasn’t a SPT to birch pollen performed? Also, can the authors explain why the SPT result to soy was only slightly positive? Especially because the patient has had a history of anaphylactic shock and positive sIgE values to soy. Did the authors consider performing oral food challenges to soy for this patient? Please also provide the SPT value – how many mms?
Answer: An SPT to birch was not performed because birch allergy was evaluated via DPA-Dx, with the results showing patient was only positive to Bet v 1 among the birch allergens tested (Bet v 2, Bet 4 and Bet v 6 were all negative). The results of the SPT (soy) may have been influenced by the condition of atopic eczema for which she takes daily corticoids and occasional histamines. It has been reported that patients with sensitive skin or dermatographism may have altered skin results (e.g.: small wheals), which may explain the low reaction to soy compared to the sIgE values [Stukus DR, et al (2016); doi: 10.1007/s11882-016-0611-z]. This possible explanation for a discrepancy between the SPT and sIgE for soy is now explained in the manuscript (lines 124-125). Following the Food Allergy Practice Parameter published in 2014, an OFC was deferred because the patient had an unequivocal and convincing clinical reactivity to soy plus a positive sIgE result; thus, a diagnostic OFC can be deferred due to the high probability of a reaction [Sampson HA, et al (2014); doi:10.1016/j.jaci.2014.05.013]. This reasoning is now stated in the manuscript as well (lines 134-135).
Comment #3: It is unclear what the authors meant by ‘birch pollen-associated clinically relevant soy allergic’. A clear definition of this should be provided in the main text. Please present the patient’s test results and symptoms against this definition. At the moment, it is a bit confusing as many different tests and symptoms were performed/reported, and at different times.
Answer: The reviewer made a good point in pointing out that the term ‘birch pollen-associated clinically relevant soy allergic’ could create confusion among readers and since it is not an established term in clinical allergy, we decided not to use this term. The allergy of the patient is now explained as a ‘conventional’ soy allergy (Gly m 5/Gly m 6) (line 55), which is a known and established term, and in the case of this patient is supported by the sIgE, BAT and ImmunoCAP Inhibition results (lines 179-182).
Comment #4: The patient is also peanut allergic. Did the authors check if the patient was also sensitised to peanut allergens that are not cross-reactive to birch pollen?
Answer: We performed an extensive peanut allergy panel via DPA-Dx: Ara h 1, Ara h 2, Ara h 3, Ara h 6 and Ara h 7 were all positive, this information is now provided in Supplemental Table 1.

Reviewer 2 Report
This paper explores the case of a patient who experiences gastrointestinal symptoms and a difference in soy tolerance during the birch pollen season. My evaluation of this paper is positive. The title and abstract provide an adequate summary of the paper, the results are clearly presented, and the tables and figures are well-designed and easy to understand.
Comment 1: The authors reported a discrepancy between SPT and pollen-specific IgE results. It is unclear why this occurred. Please clarify if any known factors can lead to such discrepancies and if any were present in the study. Are there cases where a patient has a positive pollen-specific IgE test but a negative SPT or vice versa? Additionally, please provide any hypotheses regarding why the SPT did not show a positive reaction to birch pollen despite the positive pollen-specific IgE result. Clarification on this issue would improve our understanding of allergy testing and diagnosis.
Comment 2: The results did not explicitly state that the patient tested positive for soy storage proteins Gly m 5 and Gly m 6. However, in the last statement of the first paragraph of the discussion, it was claimed that the patient had a positive SPT to these specific soy storage proteins. Can you please provide additional information from the results section that supports this claim?
Comment 3: Around 65% of the literature or references used in this study were published over 5 years ago. It is recommended to include more recent sources to ensure the study reflects the most current knowledge in the field.
Comment 4: Would it be possible to include a discussion of the limitations of the study in the conclusion section? Additionally, could you please highlight the uniqueness and significance of the case and its contribution to the current body of knowledge? This would provide a clearer understanding of the implications of your findings for future research and clinical practice.
Author Response
Dear reviewer:
We would like to thank the reviewer for the comments and suggestions, we hope we have revised the manuscript accordingly and to your satisfaction. Please note that the line numbers referred to the no-mark up version, since the version with all mark-up can alter the line numbers.
This paper explores the case of a patient who experiences gastrointestinal symptoms and a difference in soy tolerance during the birch pollen season. My evaluation of this paper is positive. The title and abstract provide an adequate summary of the paper, the results are clearly presented, and the tables and figures are well-designed and easy to understand.
Answer: We would like to thank the reviewer for this positive evaluation regarding our manuscript.
Comment 1: The authors reported a discrepancy between SPT and pollen-specific IgE results. It is unclear why this occurred. Please clarify if any known factors can lead to such discrepancies and if any were present in the study. Are there cases where a patient has a positive pollen-specific IgE test but a negative SPT or vice versa? Additionally, please provide any hypotheses regarding why the SPT did not show a positive reaction to birch pollen despite the positive pollen-specific IgE result. Clarification on this issue would improve our understanding of allergy testing and diagnosis.
Answer: An SPT to birch was not performed because birch allergy was evaluated via DPA-Dx, with the results showing patient was only positive to Bet v 1 among the birch allergens tested (Bet v 2, Bet 4 and Bet v 6). The results of the SPT may have influenced by the condition of atopic eczema for which she takes daily corticoids and occasional histamines. It has been reported that patients with sensitive skin or dermatographism may have altered skin results (e.g.: small wheals), which may explain the low reaction to soy compared to the sIgE values [Stukus DR, et al (2016); doi: 10.1007/s11882-016-0611-z]. This possible explanation for a discrepancy between the SPT and sIgE for soy is explained in the manuscript (lines 124-125).
Comment 2: The results did not explicitly state that the patient tested positive for soy storage proteins Gly m 5 and Gly m 6. However, in the last statement of the first paragraph of the discussion, it was claimed that the patient had a positive SPT to these specific soy storage proteins. Can you please provide additional information from the results section that supports this claim?
Answer: We would like to thank the reviewers for pointing this out, the three cited tested to state the patient’s Gly m 5 and Gly m 5 positivity are: sIgE, BAT and ImmunoCAP Inhibition assay, not SPT (lines 179-182). This has now been corrected.
Comment 3: Around 65% of the literature or references used in this study were published over 5 years ago. It is recommended to include more recent sources to ensure the study reflects the most current knowledge in the field.
Answer: This lack of current literature is due to two reasons (i) the majority of the research on birch seasonal variability of sIgE and clinical reactivity in food allergy is done only in birch-pollen related food; thus, Gly m 4 (ii) the research on allergy sensitization linked to seasonal variability is quite scarce. That is why, despite performing several literature researchers, not many studies were found that could be included in this manuscript. We hope that this case report brings attention to the fact that the over-reactive immune system responsible for eliciting more symptoms during the birch pollen season is not limited to patients whose food allergy is mainly driven by cross-reactivity to birch pollen allergens.
Comment 4: Would it be possible to include a discussion of the limitations of the study in the conclusion section? Additionally, could you please highlight the uniqueness and significance of the case and its contribution to the current body of knowledge? This would provide a clearer understanding of the implications of your findings for future research and clinical practice.
Answer: In the discussion, the decision of not performing a BAT against Gly m 4 and thus not evaluating clinical reactivity to Gly m 4 is explained (lines 183-185). At the end of the discussion the limitations and advantages of a case report are discussed (lines 233-244). Moreover, along the discussion and conclusion the uniqueness and implications of this case is explained as well as in the abstract. Additionally, advice for future research is provided and implications this case could have in the current clinical practice is mentioned as well.

Reviewer 3 Report
Dear Author,
It is an honor for me to be able to review this paper. This paper describes some common terms in allergology such as cross-reaction and sensitization. Anyhow, this paper could be improved in several aspects.
- please mention at the end of the introduction the novelty/rarity of this case, so that you can report this as a case report, not a case series or an observational study
- it is interesting to mention that this phenomenon is reversible? from the two examinations, we can see that there is an increase in sIgE between non-birch-pollen season and birch pollen season, but we do not know if this number goes back to normal once she is out of birch-pollen season again
- table 1 title should be modified to describe the clearer description of this table, and the current title should be put as a footnote below the table instead
- Do you also check the sIgE at the start of the birch pollen season (as shown in BAT result (Figure 1. C))? it might be interesting to see the gradual increase
- Why don't you also do a serial examination for peanut allergen (Ara h 1-2-3) out and during birch pollen season?
- Please elaborate on this statement in the "Patient Sensitization Pattern" subsection, since the table also shows that all examinations were positive with all allergens either off pollen season or peak pollen season, (above the ref. value)
"During the peak of the birch pollen season, the patient showed positive sIgE values against whole soybean and multiple soy allergens: Gly m 4, Gly m 5, and Gly m 6 and a positive value to the major birch pollen allergen Bet v 1 (Table 1)."
- please elaborate on this sentence "A patient that appears not to qualify as birch pollen-associated clinically relevant soy allergic"
- please evaluate the naming of the figure citation
"The results showed no inhibition for Gly m 4 for raw SPE and the two forms of processed soy (Figure 2A); therefore, Gly m 4 was either not present or at a very low concentration in the soy extracts. For Gly m 5 and Gly m 6, a strong inhibition signal was observed (Figure 3B and 3C), indicating that the positive BAT results with soy extract are due to the patient’s reactivity to Gly m 5 and Gly m 6."
- please also highlight if there is a difference in the usage of antihistamines and other drugs, between the off-pollen season and the peak-pollen season.
Author Response
Dear reviewer:
We would like to thank the reviewer for the comments and suggestions, we hope we have revised the manuscript accordingly and to your satisfaction. Please note that the line numbers referred to the no-mark up version, since the version with all mark-up can alter the line numbers.
It is an honor for me to be able to review this paper. This paper describes some common terms in allergology such as cross-reaction and sensitization. Anyhow, this paper could be improved in several aspects
Answer: We would like to thank the reviewer for this positive evaluation regarding our manuscript and hope that the changes implemented after this revision process greatly improves this paper.
- please mention at the end of the introduction the novelty/rarity of this case, so that you can report this as a case report, not a case series or an observational study
Answer: In the manuscript is pointed out several times that this is a case report, a section has been added in the discussion section addressing the limitations of presenting a case report ( 233-234 lines). At the same time, the advantages of a case report is the presenting novel and unique clinical cases, which is also highlighted ( 234-235 lines). The novelty of this case and the evaluation of the seasonality clinical presentation has been made clearer in the discussion (235-238 lines)
- it is interesting to mention that this phenomenon is reversible? from the two examinations, we can see that there is an increase in sIgE between non-birch-pollen season and birch pollen season, but we do not know if this number goes back to normal once she is out of birch-pollen season again
Answer: We would like to thank the reviewer for highlighting this very interesting point, we have now included this hypothesis in the discussion and made it a point that for future research, patients need to be followed for at least two years to prove indeed if this phenomenon is reversible (lines 241-244), which is what would be expected since oral tolerance changes every season.
- table 1 title should be modified to describe the clearer description of this table, and the current title should be put as a footnote below the table instead
Answer: We would like to thank the reviewer for this suggestion, the new table title is: Patient sIgE values measured out and during the peak of the birch pollen season plus soy diet restrictions and reported patient symptoms.
- Do you also check the sIgE at the start of the birch pollen season (as shown in BAT result (Figure 1. C))? it might be interesting to see the gradual increase
Answer: The decision was made to measure sIgE out and at the peak of the birch pollen season; however, this case report is considered the initial step for further research into the sensitization changes in soy allergic patients (not limited to birch pollen related, Gly m 4) according to seasonality. Therefore, a long-term prospective study where patients are followed for a at least two years is needed and repeated sIgE measurements will be needed (out, at the start and in the peak of the birch pollen season) in the future.
- Why don't you also do a serial examination for peanut allergen (Ara h 1-2-3) out and during birch pollen season?
Answer: Mainly for two reasons: (i) There was no oral tolerance variability linked to seasonality in relation to peanut, the patient can’t consume peanuts year-round (since childhood); (ii) The tested peanut allergens homologous to birch allergens were negative.
- Please elaborate on this statement in the "Patient Sensitization Pattern" subsection, since the table also shows that all examinations were positive with all allergens either off pollen season or peak pollen season, (above the ref. value) "During the peak of the birch pollen season, the patient showed positive sIgE values against whole soybean and multiple soy allergens: Gly m 4, Gly m 5, and Gly m 6 and a positive value to the major birch pollen allergen Bet v 1 (Table 1)."
Answer: The section is now titled SPT and sIgE results
- please elaborate on this sentence "A patient that appears not to qualify as birch pollen-associated clinically relevant soy allergic"
Answer: The reviewer made a good point in pointing out that the term ‘birch pollen-associated clinically relevant soy allergic’ could create confusion among readers and since it is not an established term in clinical allergy, we decided not to use this term. The allergy of the patient is now explained as a ‘conventional’ soy allergy (Gly m 5/Gly m 6), which is a known term, and in the case of this patient is supported by the sIgE, BAT and ImmunoCAP Inhibition results (lines 179-182).
- please evaluate the naming of the figure citation "The results showed no inhibition for Gly m 4 for raw SPE and the two forms of processed soy (Figure 2A); therefore, Gly m 4 was either not present or at a very low concentration in the soy extracts. For Gly m 5 and Gly m 6, a strong inhibition signal was observed (Figure 3B and 3C), indicating that the positive BAT results with soy extract are due to the patient’s reactivity to Gly m 5 and Gly m 6."
Answer: The legend of the figure has been corrected, the main title is now: “BAT results for soy and peanut allergens plus measurements out, at the start and in the peak of the birch pollen season” and the panels legends read:
- BAT results for soy allergens Gly m 5 and Gly m 6 at an allergen concentration of 300 ng/ml
- BAT results for peanut allergen Ara h 2 at an allergen concentration of 40 ng/ml
- BAT results against raw soy extract at an allergen concentration of 300 ng/mL; Period 1: December, out of the birch pollen season (P1 - color blue), Period 2: March, at the start of the birch pollen season (P2 - color red) and Period 3: May, at the peak of the birch pollen season (P3 - color green)
- please also highlight if there is a difference in the usage of antihistamines and other drugs, between the off-pollen season and the peak-pollen season.
Answer: The patient was under intermittent medication with antihistamines orally, nasally and/or as eye drops during the birch pollen season; additionally, she is on daily inhaled corticoid medication for her asthma; difference of PRN medication off and in the peak of the birch pollen season was not quantified. However, to avoid an histamine effect with the measured results of BAT, the patient was asked to stop all medications a week before each blood collection (lines 141-142).

Reviewer 4 Report
This study measured sIgE values for major soy and birch pollen allergens in a patient with soy and pollen allergies during and outside of the birch pollen season. The results suggest that the patient with soy allergy may experience worsening gastrointestinal symptoms during the birch pollen season, possibly due to an increase in IgE receptors, an overreactive immune system, and/or significant intestinal allergic inflammation.
Critique:
Although this article provides a detailed description of the case, there are still some areas that need to be modified.
1. The abstract is too lengthy and should be revised into a concise paragraph that summarizes the entire paper in simpler language.
2. The text in Figures 1 and 3 should be enlarged for better readability.
3. The font format in Table 1 needs to be unified, as Gly m 5 is in bold while Gly m 6 is not, and "As =. Anaphylactic shock" should have the "." removed.
4. As this article presents the results of a single case study, it should be considered as preliminary findings, and the author should clarify whether these observations apply to a broader population.
Author Response
Dear reviewer:
We would like to thank the reviewer for the comments and suggestions, we hope we have revised the manuscript accordingly and to your satisfaction. Please note that the line numbers referred to the no-mark up version, since the version with all mark-up can alter the line numbers.
Critique:
Although this article provides a detailed description of the case, there are still some areas that need to be modified.
1. The abstract is too lengthy and should be revised into a concise paragraph that summarizes the entire paper in simpler language.
Answer: The abstract has been shortened (Before: 381 words; Now: 275 words) and the main message is now clearer to readers.
The text in Figures 1 and 3 should be enlarged for better readability.
Answer: Thank you for the suggestion, this changes have been made to text in both figures
The font format in Table 1 needs to be unified, as Gly m 5 is in bold while Gly m 6 is not, and "As =. Anaphylactic shock" should have the "." removed.
Answer: Thank you for pointing out this, the table font and designed is now unified and the misplaced “.” has been removed
As this article presents the results of a single case study, it should be considered as preliminary findings, and the author should clarify whether these observations apply to a broader population.
Answer: This has been clarified both in the abstract and in the discussion (lines 233-235) and the conclusion.

Round 2
Reviewer 3 Report
There is a very good improvement done by the author,
However, I noticed that the aim was removed from the manuscript which I think is still important, otherwise, everything is already good.
Reviewer 4 Report
Thank you for addressing my concerns. Your efforts have resulted in a current version that effectively and clearly presents your research.